# The Nitrogen Cycling Key Functional Genes and Related Microbial Bacterial Community α−Diversity Is Determined by Crop Rotation Plans in the Loess Plateau

**Rui Liu [1], Yang Liu [2], Yuan Gao [2], Fazhu Zhao [2] and Jun Wang [1,2,*]**

[1] State Key Laboratory of Soil Erosion and Dryland Farming on the Loess Plateau, Institute of Soil and Water Conservation, Chinese Academy of Sciences and Ministry of Water Resources, Yangling 712100, China; liurui216@mails.ucas.ac.cn

[2] Shaanxi Key Laboratory of Earth Surface System and Environmental Carrying Capacity, College of Urban and Environmental Science, Northwest University, Xi'an 710127, China

[*] Correspondence: wangj@nwu.edu.cn; Tel.: +86-29-8830-8783

**Abstract:** Soil nitrogen cycling microbial communities and functional gene α−diversity indicate soil nitrogen cycling ecological functions and potentials. Crop rotation plans affect soil nitrogen fractions and these indicators. We sequenced soil samples from four crop rotation plans (fallow, winter wheat monoculture, pea-winter wheat-winter wheat-millet rotation, and corn-wheat-wheat-millet rotation) in a long-term field experiment. We examined how microbial communities and functional gene α−diversity changed with soil nitrogen fractions and how nitrogen fractions regulated them. Planting crops increased the abundance and richness of nitrogen cycling key functional genes and bacterial communities compared with fallow. The abundance and richness correlated positively with nitrogen fractions, while Shannon index did not. The abundance increased with soil total nitrogen (STN) and potential nitrogen mineralization (PNM), while Shannon index showed that nitrogen cycling key functional genes increased and then decreased with increasing STN and PON. Introducing legumes into the rotation improved the α−diversity of nitrogen cycling key functional genes. These results can guide sustainable agriculture in the Loess Plateau and clarify the relationship between nitrogen fractions and nitrogen cycling key functional genes.

**Keywords:** nitrogen cycling genes; nitrogen fractions; microbial community; crop rotation; loess Plateau

## 1. Introduction

The abundance of functional genes is an important indicator for evaluating ecosystem functionality [1]. Nitrogen cycling functional genes, including amoA, amoB, and amoC for nitrification, nirS, nirK, and nosZ for denitrification, and ureC for ammonification, have been applied to gauge the nitrogen transformation capacity in agricultural soils at varying stages [2–4]. Several studies have demonstrated that agricultural management practices alter functional gene abundance, for instance, reducing tillage or implementing no-till practices, which increase the abundance of amoA, nirK, and nosZ [5–7]. Additionally, the application of organic fertilizers can boost the abundance of ureC, amoA, nirS, nirK, and nxrB in nitrogen cycling [8,9]. In recent years, functional gene diversity has also evolved as an essential factor for assessing the functional potential of soil microbial communities in ecosystems [10,11]. Researchers have explored the functional gene diversity in different ecosystems, such as grasslands, forests, and agricultural lands, to obtain knowledge on the functional potential of soil microbial communities [12–14]. Moreover, Chukwuneme et al. [15] used metagenomic sequencing to investigate the impact of different agricultural management practices on the diversity of carbon cycle genes in the corn rhizosphere soils, while a meta-analysis by You et al. [16] revealed that nitrogen addition affects the diversity of crucial functional genes linked to $N_2O$ emission in agricultural

ecosystems. Hence, exploring the abundance and diversity of functional genes is essential in understanding the ecological functions and potential of soil microbial communities.

The Loess Plateau is facing severe soil nutrient deficiency due to long-term soil erosion. To address this issue, crop rotation has been widely used to enhance the soil nitrogen pool [17,18]. Fu et al. [19] conducted research indicating that long-term diversified crop rotation can effectively improve soil nitrogen fractions and sustain dryland planting regimes. The corn-legume-wheat rotation, for example, can increase soil particulate organic nitrogen and mineralized nitrogen [20], while rice-rape rotation can increase soil particulate organic nitrogen and microbial nitrogen [21]. Additionally, crop rotation can improve soil structure, the living environment for microorganisms, and increase microbial diversity [22]. Additionally, long-term crop rotation can increase crop diversity, and the input of diversified crop residues and litter can result in changes to the substrate available to soil microbes, leading to an increase in microbial $\alpha-$diversity [23–25]. Nonetheless, studies suggest that long-term crop rotation may lead to the continuous accumulation of high-C/N crop residues in the soil [26], which could result in a decrease in microbial diversity [27]. Therefore, introducing crops with different C/N ratios, such as legumes or non-legumes, into long-term crop rotation plans may have varying effects on microbial $\alpha-$diversity.

Changes in soil microbial communities can result in alterations in functional genes involved in nitrogen cycling [28,29]. Several studies have demonstrated that crop rotation can regulate the abundance of denitrification genes (nirK, nosZ, and nirS), thereby influencing $N_2O$ emissions [30,31]. However, soil nitrogen fractions, as important factors reflecting soil productivity and nitrogen status [32], can also affect changes in nitrogen cycling functional genes [4,33], but there are few studies in this direction. In crop rotation plan, different crops have varied litter properties that can impact soil nutrient content [22]. Legumes, for instance, release nutrient-rich, juicy, and protein- and sugar-rich residues, which result in faster nutrient release than the more stable organic matter derived from the fibrous plant residues of crops such as corn and cereals [34]. Therefore, further research is necessary to assess the impact of nitrogen fractions in different long-term crop rotation plans on functional gene $\alpha-$diversity.

This study aims to investigate the impact of different planting plans on soil nitrogen fractions, key functional genes and related microbial communities' $\alpha-$diversity through 36 years of field experiments. It is hypothesized that: (1) crop rotation plans can increase the abundance of key functional genes for nitrogen cycling and the richness of related microbial communities; (2) long-term crop residue from rotation plans may result in differences in nitrogen cycling key functional genes and related microbial communities' diversity due to varying C/N ratios. The objectives of this study are to elucidate the effects of different crop rotation plans on key functional genes and related microbial communities involved in nitrogen cycling and to explore the regulatory modes of soil nitrogen fractions on key functional genes $\alpha-$diversity.

## 2. Materials and Methods

### 2.1. Experimental Site and Treatments

A long-term field experiment was established in September 1984 at the Changwu Agroecological Station (35°12′ N, 107°44′ E) in Changwu County, Shaanxi Province, China. The study area is representative of a typical rainfed farming system in the Loess Plateau of China. The experimental site has a continental monsoon climate, with a mean annual temperature of 9.1 °C and a frost-free period of 194 d. The long-term (1984–2020) average annual precipitation is 580 mm, half of which occurs from July to September. The soil is a Heilutu silt loam (Calcarid Regosol according to the FAO classification system or Ultisol according to the U.S. soil taxonomy), with 45 g kg$^{-1}$ sand, 656 g kg$^{-1}$ silt, 309 g kg$^{-1}$ clay, 8.4 pH, 105 g kg$^{-1}$ CaCO$_3$, 10.5 g kg$^{-1}$ organic C, 1.0 g kg$^{-1}$ total N, and 1.4 Mg m$^{-3}$ bulk density at the 0–30 cm depth at the beginning of the experiment.

There were four cropping designs for the study, including (1) fallow (F); (2) winter wheat monoculture (W); (3) 1 year of pea, 1 year of winter wheat, and 1 year of winter wheat

and millet (PWWM); (4) 1 year of corn, 1 year of winter wheat, and 1 year of winter wheat and millet (CWWM). A randomized block design was used in this experiment with three replications. Each plot had 10.3 m by 6.5 m size separated by 0.5 m strip, and each block was separated by 1 m strips. Crops were planted by hand under conventional tillage using animal-drawn (first 16 yr) and hand tractor-drawn (second 18 yr) plows to a depth of 10 cm. Crops were planted at 20 cm row spacing, except for corn which was planted at 70 cm spacing. Plant populations were 2.23, 0.60 and 0.04 million plants ha$^{-1}$ for winter wheat, pea and corn, respectively, and the seeding rate was 28 kg ha$^{-1}$ for millet, respectively. At planting, chemical fertilizers were broadcasted to winter wheat, corn and millet using urea (46% N) and monoammonium phosphate (11% N, 23% P) at rates of 120 kg N ha$^{-1}$ and 20 kg P ha$^{-1}$. Pea received N and P from monoammonium phosphate at 10 kg N ha$^{-1}$ and 20 kg P ha$^{-1}$, respectively. Because of the high soil potassium content, no potassium fertilizer was applied. Weed management was carried out by hand before, during, and after crop growth. Pesticides were applied as needed to control pests.

### 2.2. Soil Samplings

In September 2020, samples were collected after peas and corn were harvested and before wheat was planted. Soil samples were collected from a depth of 15 cm from three places in central rows of each plot using a hand probe (5 cm inside diameter) in September 2020. A separate undisturbed soil core (5 cm inside diameter) was collected simultaneously at 0–15 from each plot for the bulk density. The collected soil samples were immediately stored in sterile plastic bags, placed in iceboxes, and brought back to the laboratory immediately. Then, all the samples were sifted through a 2 mm mesh, crop residues, root materials, and stones were removed. and were thoroughly homogenized to be further divided into three parts: one part was air-dried to analyze the nitrogen fractions, another part was stored at $-4\,^\circ$C for determination of MBN (microbial biomass nitrogen), while the rest of the samples were stored at $-80\,^\circ$C until DNA extraction and metagenomic sequencing.

### 2.3. Analyses of Soil Nitrogen Fractions

The soil total nitrogen (STN) concentration was determined by the combustion method using a high induction furnace N analyzer (Euro Vector EA3000, Manzoni, Italy) [35]. The particulate organic nitrogen (PON) was determined with the sodium hexametate separation method [36]. The potential nitrogen mineralization (PNM) concentration was determined by the incubation method modified by Haney et al. [37]. The NH$_4$$^+$-N and NO$_3$$^-$-N concentrations in the extract were determined using the modified Griess-Ilosvay method with an autoanalyzer (Lachat Instruments, Loveland, CO) [35]. The soil water content (SWC) and bulk density (BD) were measured from the gravimetric weight of the core before and after oven drying at 105 $^\circ$C for 24 h. The other container with moist soil was subsequently used for determining MBN concentration using the modified fumigation-incubation method for airdried soils [38].

### 2.4. DNA Extraction, Sequencing, and Data Processing

Total genomic DNA was extracted from 0.5 g soil samples using the E.Z.N.A.$^\circledR$ Soil DNA Kit (Omega Bio-tek, Norcross, GA, USA) according to manufacturer's instructions. Concentration and purity of extracted DNA was determined using TBS-380 and NanoDrop2000, respectively. DNA extract quality was checked on 1% agarose gel. There were three replicates for each soil sample to obtain sufficient DNA for shotgun metagenome sequencing. The metagenome was sequenced in an Illumina HiSeq 2000 platform (Personal, Shanghai, China) to generate 150 bp paired-end reads at a greater sequencing depth. Reads aligned to the human and vegetation genome were removed, and the lengths were trimmed using Sickle. Sequence data associated with this project have been deposited in the NCBI Short Read Archive database (Accession Number: PRJNA985043; https://www.ncbi.nlm.nih.gov/sra/PRJNA985043 accessed on 4 June 2022).

### 2.5. Metagenomic Analysis

To enhance the reliability and quality of subsequent analysis, we removed adapter sequences, and discarded those quality-trimmed reads of less than 50 bp in length or containing N (ambiguous) bases [39]. Megahit software (https://hku-bal.github.io/megabox/, accessed on 15 April 2021) was used to obtain high-quality reads [40], MetaGeneMark (http://exon.gatech.edu/GeneMark/metagenome, accessed on 20 April 2021) was used to predict genes in contigs longer than 200 bp [41]. Gene abundance for each sample was the trans per million values [TPM: (Reads Number/Gene Length)_Relative] × 1,000,000] [42].

According to the results of the KEGG database search, functional annotation and taxonomic assignment of each sample were carried out for further analysis using eggNOG-mapper v2, a tool based on precomputed orthology assignments. Based on a previous study, 17 N-cycling functional genes were defined as nitrogen cycling genes, associated with the processes of Nitrification, Denitrification, Dissimilatory Nitrate Reduction (DNR), Assimilatory Nitrate Reduction (ANR), Ammoniation, and Assimilation [43]. Detailed information on N-cycling functional genes is listed in Table S1.

### 2.6. Screening Key N-Cycling Functional Genes

We screened key N-cycling functional genes using network analysis and random forests [44]. Firstly, the 17 N-cycling functional genes were screened based on the KEGG databases. To explore the relationships among N-cycling functional genes, Spearman correlation analyses were performed using the "Hmisc" and "igraph" packages [45], genes with a relative abundance higher than 0.1% were kept for network construction. A valid co-occurrence was considered a statistically robust correlation ($|r| > 0.6$, $p < 0.05$). The network was visualized by Gephi platform [46]. We combined the overall samples and used the network analyses to select 11 genes with respect to hub nodes in network, including amoA, amoC, napA, nirK, nirS, norB, nasA, nasB, nirA, ureC and gdh (Supplementary Materials Figure S1A) according to degree (degree > 0). Secondly, we used random forests to disentangle the contributions of N-cycling functional genes to variations in different nitrogen fractions (Supplementary Materials Figure S1B) using the 'randomforest' package [47]. Finally, on the basis of the results of co-occurrence network and random forest analyses, 6 genes, including amoC, nirS, norB, nasA, nasB, gdh, were selected as the key N-cycling functional genes in our study.

### 2.7. Statistical Analyses

All statistical analyses were conducted in the R environment (v4.0.5; http://www.r-project.org/, accessed on 30 April 2021). One-way analysis of variance (ANOVA) was performed based on the 'stats' package to assess the effect of different crop rotation plans on nitrogen fractions content, abundance and the Shannon index of N-cycling genes, and abundance and Shannon index of microbial functional community at the 0.05 level of significance. Diversity of Shannon index and richness index (Chao1, ACE) were calculated using "phyloseq" package [48]. Nonlinear regression and spearman correlation analyses were performed using the "survival" and "basicTrendline" packages, respectively [49]. Heatmaps were used to illustrate the Z-score-normalized relative abundance of N-cycling functional genes using the "pheatmap" package [50]. Other graphs were drawn using the "ggplot2" package [51].

## 3. Results

### 3.1. Soil Nitrogen Fractions

Different crop rotation plans have been found to affect the nitrogen fractions in soil (Figure 1). Compared with F, other cropping plans significantly increased the content of total nitrogen (STN) and potential mineralizable nitrogen (PNM) in the soil. Compared with W, CWWM significantly increased STN by 23.9%, while PNM decreased by 12.7% ($p < 0.05$); Compared with W, PWWM or CWWM significantly increased the content of microbial nitrogen (MBN) by 47.4% and 60.3%, respectively ($p < 0.05$). Compared with

PWWM, CWWM increased STN by 28.6% and particulate organic nitrogen (PON) by 88.2% ($p < 0.05$). In addition, the content of ammonium nitrogen and nitrate nitrogen in the soil did not show significant differences in different crop rotation plans.

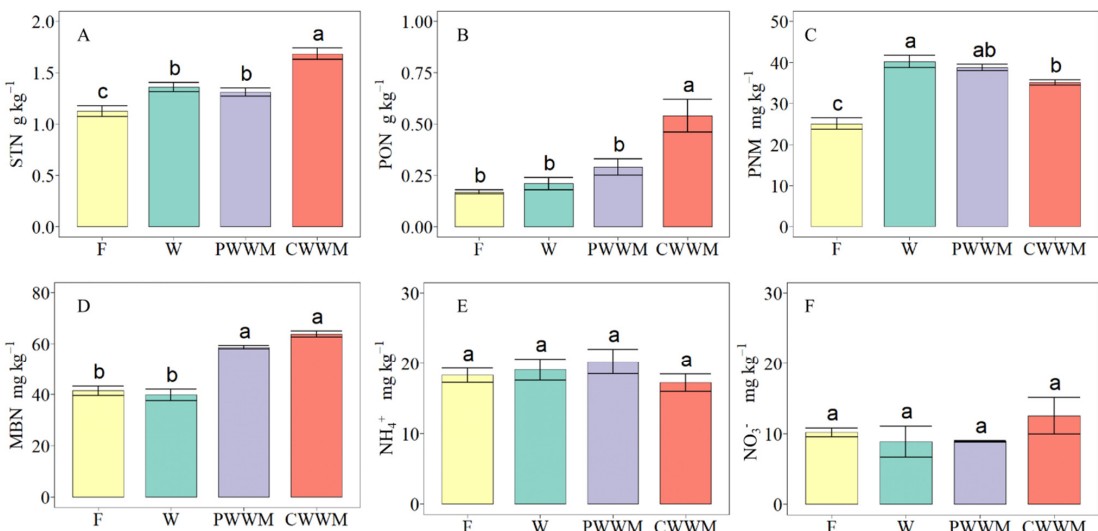

**Figure 1.** Effects of different crop rotation plans on soil nitrogen fractions. Soil N fractions including (**A**) STN, soil total nitrogen; (**B**) PON, particulate organic nitrogen; (**C**) PNM, potential nitrogen mineralization; (**D**) MBN, microbial biomass N; (**E**) $NH_4^+$, ammonium nitrogen; (**F**) $NO_3^-$, nitrate nitrogen. Mean values ± standard deviation of nitrogen fractions. Different small letters indicate significant differences among crop rotation plans ($p < 0.05$, ANOVA, Tukey HSD).

### 3.2. Alpha Diversity of Key Function Genes in Nitrogen Cycling

There were significant differences in the abundance and $\alpha-$diversity of key functional genes under different crop rotation plans in our investigation. The abundance of key functional genes varied significantly in different crop rotation plans. The abundance of amoC and nasA in F was significantly lower than in other treatments. Moreover, the abundance of norB and nasB in F was significantly lower than in PWWM. PWWM had a significantly higher abundance of nasB than CWWM. In addition, there was no significant difference in the abundance of nirS and gdh among different crop rotation plans (Table 1). Furthermore, compared with W, PWWM and CWWM significantly increased the abundance of amoA and nirK (Table S1). The abundance and $\alpha-$diversity of key functional genes also differed under different crop rotation plans. The total abundance of key functional genes in F was significantly lower than in other treatments (Figure 2A). There was no significant difference in the total abundance of key functional genes among PWWM, CWWM and W. The Shannon of F was significantly lower than that of PWWM (Figure 2B); the Shannon of W and PWWM was significantly higher than that of CWWM; there was no significant difference in Shannon between PWWM and W.

**Table 1.** Effects of different crop rotation plans on the abundance of nitrogen cycle key genes values.

| Processes | Nitrification | Denitrification | | ANR (Assimilatory Nitrogen Reduction) | | Assimilation |
|---|---|---|---|---|---|---|
| **Crop Rotation Planss** | **amoC** | **norB** | **nirS** | **nasA** | **nasB** | **gdh** |
| | $(NH_4^+ \rightarrow NH_2OH)$ | $(NO \rightarrow N_2O)$ | $(NO_2^- \rightarrow NO)$ | $(NO_3^- \rightarrow NO_2^-)$ | $(NO_3^- \rightarrow NO_2^-)$ | $(NH_4^+ \rightarrow Org)$ |
| F | 124.67 ± 7.33 b | 714.67 ± 18.56 b | 57.33 ± 8.11 a | 1149.33 ± 53.07 b | 99.33 ± 10.73 bc | 1720.67 ± 70.70 a |
| W | 184.00 ± 7.57 a | 753.33 ± 20.80 ab | 39.33 ± 11.79 a | 1351.33 ± 28.99 a | 127.33 ± 5.81 ab | 1906.67 ± 75.10 a |
| PWWM | 174.00 ± 9.17 a | 858.67 ± 20.34 a | 44.67 ± 2.40 a | 1362.67 ± 35.63 a | 154.67 ± 10.48 a | 1960.67 ± 16.18 a |
| CWWM | 170.00 ± 2.00 a | 788.00 ± 69.41 ab | 56.00 ± 2.00 a | 1473.33 ± 34.57 a | 88.00 ± 7.21 c | 2004.67 ± 134.39 a |

Note: F, fallow; W, wheat monoculture; PWWM, pea-wheat-wheat-millet rotation; CWWM, corn-wheat-wheat-millet rotation. Values within the same column followed by different letters indicate significant differences ($p < 0.05$, ANOVA, Tukey HSD). are mean ± standard deviation ($n = 3$).

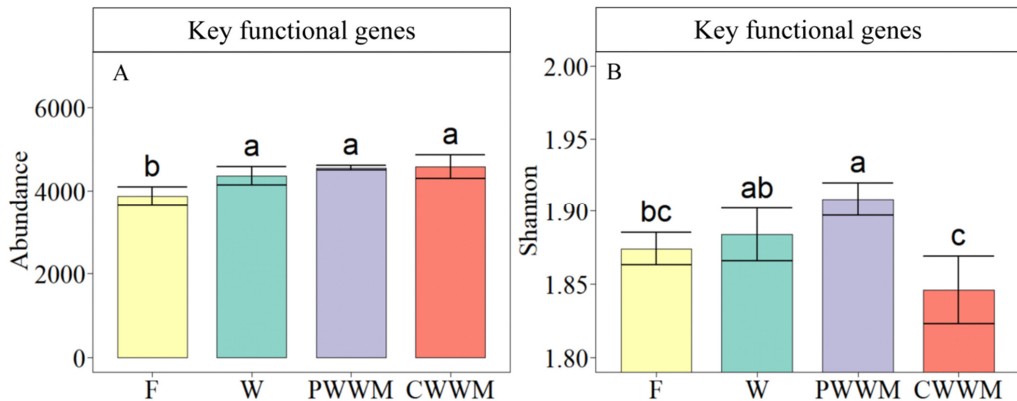

**Figure 2.** Values of $\alpha-$diversity of N-cycling key functional genes in different crop rotation plans. (**A**) the abundance of key N-cycling functional genes; (**B**) the Shannon of key N-cycling functional genes. Mean values $\pm$ standard deviation of the abundance and the Shannon of nitrogen cycle genes. Different small letters indicate significant differences among crop rotation plans ($p < 0.05$, ANOVA, Tukey HSD).

### 3.3. Alpha Diversity of Microbial Communities Related to Key Functional Genes

In species related to key functional genes in the nitrogen cycle at the phylum level, the relative abundance of *Proteobacteria* in CWWM was higher than that in W, while the relative abundance of *Actinobacteria* was lower in CWWM than in PWWM (Figure 3; Table S3). There were no significant changes in the relative abundance of other species. However, there was no significant difference in the relative abundance of microorganisms between CWWM and PWWM at class, order, family, and genus levels (Figure S2). In this study, a total of 546 microbial species related to key functional genes in the nitrogen cycle were detected, and there were differences in bacteria $\alpha-$diversity among different crop rotation plans. Compared with the W, the Chao1 index of the rotation plans was significantly increased, and the ACE index of CWWM was significantly increased compared with the F. However, there was no significant difference in the richness index (Chao1, ACE) between PWWM and CWWM. In addition, there was no significant difference in the Shannon index of microbial communities among different crop rotation plans (Table 2).

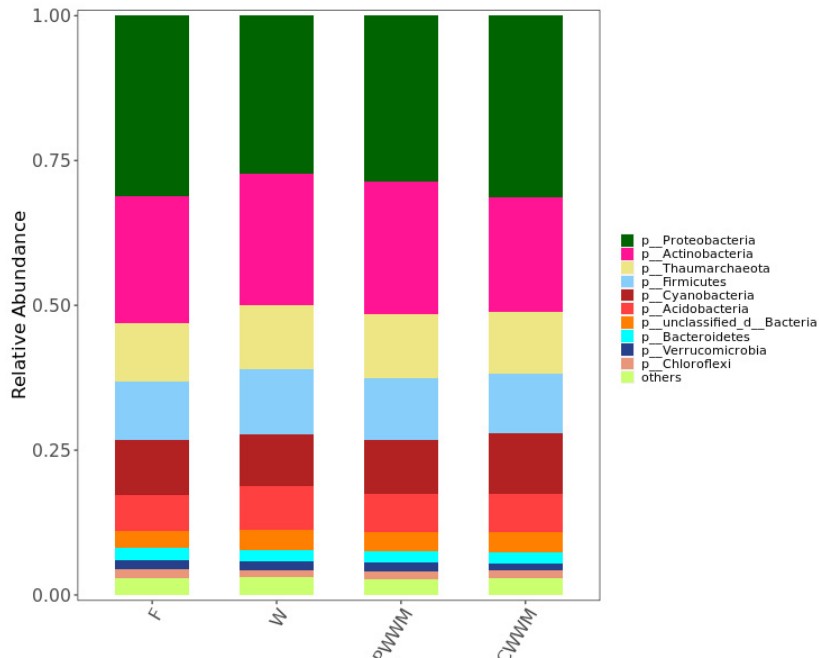

**Figure 3.** Relative abundance of the phyla of species related to key functional genes.

**Table 2.** Effects of different crop rotation plans on $\alpha$−diversity of microbial community. Values are mean ± standard deviation (*n* = 3).

| Kingdom | Bacteria | | | Archaea | | |
|---|---|---|---|---|---|---|
| Crop Rotation Planss | Shannon | Chao1 | ACE | Shannon | Chao1 | ACE |
| F | 6.94 ± 0.04 a | 297.33 ± 3.53 bc | 7.47 ± 0.12 b | 0.94 ± 0.08 a | 3.67 ± 0.67 a | 0.91 ± 0.09 a |
| W | 6.87 ± 0.08 a | 292.33 ± 8.51 c | 7.79 ± 0.13 a | 1.08 ± 0.03 a | 4.67 ± 0.33 a | 1.06 ± 0.03 a |
| PWWM | 6.89 ± 0.01 a | 313.33 ± 2.60 ab | 7.78 ± 0.06 ab | 1.00 ± 0.04 a | 3.67 ± 0.33 a | 0.89 ± 0.05 a |
| CWWM | 6.97 ± 0.05 a | 325.67 ± 4.33 a | 7.95 ± 0.02 a | 0.98 ± 0.01 a | 4.00 ± 0.58 a | 0.97 ± 0.08 a |

Note: F, fallow; W, wheat monoculture; PWWM, pea-wheat-wheat-millet rotation; CWWM, corn-wheat-wheat-millet rotation. Values within the same column followed by different letters indicate significant differences ($p < 0.05$, ANOVA, Tukey HSD). Bold values indicate significant differences.

### 3.4. The Regulatory Modes of Soil Nitrogen Fractions on Key Functional Genes and Microbial Communities

As the key functional genes that affect the changes of nitrogen fractions, their abundance changes must be positively correlated with nitrogen fractions, which was shown by Spearman correlation analysis and confirmed by regression models in this study (Table 3; Figure 4A,B). However, Spearman correlation analysis revealed that The Chao1 index of the bacteria community was significantly positively correlated with PON and MBN. Furthermore, nitrogen fractions had no significant effect on the Shannon index of the key functional genes and microbial community. In addition, the results of the regression model also showed that the Shannon index of the key functional genes increased first and then decreased with the increase in STN and PON, and higher levels of STN and PON led to a decrease in the Shannon index of the key functional genes.

**Table 3.** Spearman correlation coefficients between the N fractions and $\alpha$−diversity of microbial community.

| Crop Rotation Plans | Key Functional Genes | | Bacteria | | | Archaea | | |
|---|---|---|---|---|---|---|---|---|
| | Abundance | Shannon | Shannon | Chao1 | ACE | Shannon | Chao1 | ACE |
| STN | 0.720 ** | −0.399 | −0.05 | 0.482 | 0.557 | 0.02 | 0.177 | 0.234 |
| PON | 0.698 * | −0.407 | 0.053 | 0.583 * | 0.455 | −0.03 | 0.155 | 0.2 |
| PNM | 0.273 | 0.476 | −0.27 | 0.1 | 0.564 | 0.531 | 0.298 | 0.288 |
| MBN | 0.706 * | −0.238 | 0.146 | 0.809 ** | 0.52 | −0.249 | −0.209 | −0.231 |
| $NH_4^+$ | 0.259 | 0.42 | −0.402 | −0.212 | −0.006 | −0.039 | −0.003 | −0.164 |
| $NO_3^-$ | −0.399 | −0.448 | 0.375 | 0.365 | 0.1 | −0.225 | −0.303 | −0.284 |

Note: ** Correlation is significant at $p < 0.01$; * Correlation is significant at $p < 0.05$.

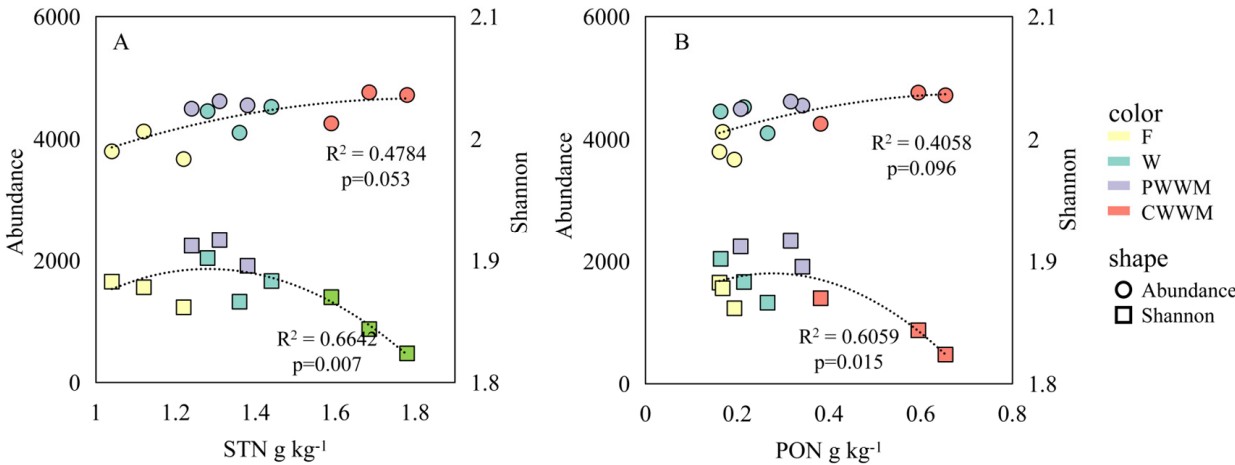

**Figure 4.** Relationships between $\alpha$−diversity of key functional genes and nitrogen fractions. STN, soil total nitrogen; PON, particulate organic nitrogen. (**A**) Relationships between $\alpha$−diversity of key functional genes and STN; (**B**) Relationships between $\alpha$−diversity of key functional genes and PON; The nonlinear fitting method is quadratic fitting. The fitting degree ($R^2$) of the curve is provided.

## 4. Discussion

### 4.1. Variations in Soil Nitrogen Fractions under Different Crop Rotation Plans

Soil nitrogen (N) plays a central role in soil quality and biogeochemical cycles [52], and is available in various chemical fractions, including soil total N (STN), particulate organic matter (PON), potential nitrogen mineralization (PNM), microbial biomass N (MBN), ammonium N ($NH_4^+$-N), and nitrate N ($NO_3^-$-N) [53]. Different nitrogen fractions could better reflect changes in soil quality and nitrogen supply potential that alter nutrient dynamics in agroecosystems [54]. Therefore, it is essential to study the transformations of nitrogen fractions under different agricultural management measures for assessing changes in soil quality and function [55].

The results showed that corn-based crop rotation significantly increased STN and PON compared to winter wheat monoculture, while pea-based crop rotation had no significant effect. However, both corn-based and pea-based crop rotations increased PNM compared to winter wheat monoculture. The higher STN and PON under corn−based crop rotation may be attributed to the quality and quantity of crop residue. Corn produces more biomass than leguminous crops, which can provide more external nitrogen input to the soil [56,57]. Moreover, corn residue has a high C/N ratio, which can slow down its decomposition rate and increase soil nitrogen accumulation [58]. On the other hand, leguminous crops have a lower C/N ratio, which can increase soil mineralization rate and PNM. This is consistent with previous studies that reported higher PNM after legume rotation than after corn rotation [59].

The increase in PNM under both corn-based and pea-based crop rotations may also be related to the symbiotic nitrogen fixation by leguminous crops. Legumes can enhance soil nitrogen fixation by establishing a symbiotic relationship with rhizobia bacteria. This can increase the availability of nitrogen for subsequent crops and improve soil fertility. Previous studies have also reported that rotation with legumes can increase PNM compared to monocropping [60]. The results of this study also indicated that crop rotation plans can increase microbial biomass nitrogen (MBN) compared to monocropping. This may be due to the increased diversity and quantity of crop residues and root exudates under rotation plans, which can provide more substrates for microbial growth and activity [61]. This finding is in agreement with other studies that reported higher MBN under rotation plans than under monocropping [62,63].

The findings of this study suggest that corn-based crop rotation can improve soil nitrogen status and fertility more than pea-based crop rotation or winter wheat monoculture. However, this study has some limitations that need to be addressed. First, the study was conducted in a single site with a specific soil type and climate condition, which may limit the generalizability of the results. Second, the study only measured soil nitrogen parameters at one time point after harvest, which may not reflect the seasonal dynamics of soil nitrogen processes.

### 4.2. Effects of Different Crop Rotation Plans on Alpha Diversity of Key Functional Genes

Environmental factors can effectively regulate the abundance of soil nitrogen cycling genes [64]. In this study, compared with F, the total abundance of key functional genes in other treatments was significantly increased, which may be related to the increased organic matter input and PNM by planting crops (Figure 1C); moreover, crop roots can also improve soil bulk density (BD) and provide a more diversified living space for microbial communities, thereby increasing microbial richness [65]. Brandan et al. [66] reported a significant negative correlation between soil BD and the abundance of nitrogen cycling functional gene in a long-term rotation study, so the higher BD in F may also be one of the reasons for its lower key functional gene abundance than other treatments (Table S2). In addition, soil pH and soil water content (SWC) are also the main factors affecting the abundance of soil nitrogen cycling genes [67,68]. However, due to the factors such as climate, tillage and rainfall in the Loess Plateau, there were no significant differences in pH, BD and SWC between monoculture and two rotation plans, which was also an important

reason for the lack of significant difference in the total abundance of key functional genes among these three treatments.

Soil functional gene diversity reflects the potential of soil microorganisms to perform various metabolic activities and respond to environmental changes, which is important for maintaining soil health and function [69]. In this study, we found that CWWM significantly reduced the Shannon index of key functional genes compared with W and PWWM, indicating that introducing maize into rotation reduced the functional potential of nitrogen cycling in the ecosystem, while introducing legumes into rotation had more positive effects on the functional potential of nitrogen cycling in the ecosystem. Soil functional gene diversity can also influence the stability and resilience of soil ecosystems, as it can buffer against disturbances and enhance functional redundancy [70]. Especially in the Loess Plateau region, introducing legumes into rotation can more effectively improve the risk resistance of soil ecosystems. We speculate that the main reasons for the difference in key gene diversity between the two different rotation systems are as follows: first, legume crops can form symbiotic relationships with specific bacteria (i.e., *Actinobacteria*) during their growth, resulting in increased abundance of specific genes (i.e., nasB) [71]; second, legume crop residues have a lower C/N, which makes them easier to decompose and provide nutrients for microorganisms faster [72]; meanwhile, in practical agricultural management, the nitrogen application rate for legume crops is only 1/6 of that for maize crops, and crop residue amount is also lower than that for maize crops, resulting in soil nitrogen deficiency, leading to a competitive relationship between microorganisms and crops, with microorganisms having an advantage [73,74]. After long-term introduction of maize into rotation, although it can increase yield and economic benefits, the risk of reducing key gene diversity should be considered. Hence, future studies should explore the combination of ecological agriculture measures, such as organic fertilization or introduction of legume into rotation, with introduction of maize into rotation to enhance the ecosystem sustainability.

*4.3. Effects of Different Crop Rotation Plans on Alpha Diversity of Microbial Communities*

The quality and quantity of crop residues may be the main driving factor affecting the composition of microbial communities [75]. Among the microbial communities related to key functional genes, the relatively high abundance of *Proteobacteria* and *Actinobacteria* changes in different crop rotation plans. The relative abundance of *Proteobacteria* in crop rotation is higher than that in single cropping, which is similar to the law of STN content. Li et al. and Wang et al. [76,77] also confirmed that the relative abundance of *Proteobacteria* is related to organic matter content because *Proteobacteria* can decompose recalcitrant carbon sources into small intermediate molecules to provide nutrition for other microorganisms [78]. Therefore, the relative abundance of *Proteobacteria* in CWWM is significantly higher than that in monoculture. However, the relative abundance of *Actinobacteria* in CWWM is significantly lower than that in PWWM and W. The refractory corn residue makes it difficult for *Actinobacteria* to grow in the soil, as *Actinobacteria* mainly use available carbon sources for growth [79]. This can also explain why there is no difference in STN and PON content between PWWM and W.

The $\alpha-$diversity of microbial communities plays an important role in ecosystem processes [80]. Numerous studies have demonstrated that crop rotation plans can effectively increase the $\alpha-$diversity of soil microbial communities [81,82]. This is because both legume-wheat and corn-wheat rotations can provide diverse substrates for microbial utilization through diverse plant residues and root secretion, respectively; Moreover, the growth of different crops can alter soil nutrient conditions and soil environment, which can directly or indirectly enhance the diversity and abundance of soil microbial communities [83–86]. In this study, crop rotation improved the richness of microbial communities but did not change the Shannon index of microbial communities. This may be due to the dominant role of long-term fertilizer inputs and mechanical cultivation in regulating soil biodiversity in farmland [87,88]. Additionally, the sampling period in this study was after crop harvest,

and the farmland soils were in a fallow state, which may have led to non-significant changes in the Shannon index of microbial communities.

### 4.4. Effects of Nitrogen Fractions on the α−Diversity of Key Functional Genes and Associated Microbial Communities

The richness of microbial communities is closely related to the quantity and quality of their available resources [89,90], with soil nitrogen content being a limiting factor for microbial growth, and soil microbial community richness increasing as soil nitrogen accumulated [91,92]. In the crop rotation plan, the input of different crop residues can increase the availability of substrates, thus increasing microbial community richness [93]. This study also showed a significant positive correlation between microbial community richness and STN, PON (Table 3). However, while exogenous input of crop residues may change soil microbial community richness, different species can have the same ecological function in an ecosystem [94], and so changes in microbial communities may not necessarily result in the corresponding changes in functional genes.

We found a significant positive correlation between the abundance of key functional genes and nitrogen fractions, but the Shannon index showed an initial increase followed by a decrease with an increase in STN and PON, which is consistent with our hypothesis. Compared with F, different agricultural cropping plans lead to an increase in soil organic matter, which will provide a food source for microorganisms participating in nutrient cycling during mineralization [95]. Different crop rotation plans change the microbial community and also regulate nitrogen cycling functional genes [96]. Therefore, in this study, the abundance of key functional genes was positively correlated with STN and PNM (Table 3). However, the change in the Shannon index of the key functional genes may be related to the nitrogen environment and crop residue quality (Figure 4). When the soil STN or PON content is low, the Shannon index of key functional genes in nitrogen cycling increases with an increase in STN, as diversified crop residue inputs in crop rotation plans will provide more ecological niches for microorganisms, thereby increasing microbial diversity [97,98]. However, when the STN or PON content is high, the Shannon index of key functional genes in nitrogen cycling decreases. Similar findings have been reported, such as a significant increase in soil organic matter and total nitrogen content in long-term organic management but a decrease in soil microbial diversity [99,100]. This may be because higher soil nitrogen provides abundant resources for microorganisms, allowing microorganisms with lower resource utilization efficiency to occupy the main ecological niche [76]. This may lead to a reduction in the diversity of key functional genes. In addition, the excessive input of high C/N ratio crop residues into the soil in the CWWM causes a disturbance in the soil C/N, affecting the metabolic activities of microorganisms towards carbon sources [93]. Therefore, although introducing corn into the wheat planting can effectively increase the soil nitrogen fractions, long-term crop residue input will have a negative impact on the potential of nitrogen cycling function in soil on the Loess Plateau.

### 5. Conclusions

Based on a 36-year field experiment in the Loess Plateau, the results showed that compared with wheat monoculture, maize rotation contributed to the accumulation of soil STN and PON, while legume rotation effectively maintained the soil STN and PON contents. Crop rotation did not change the Shannon index of microbial communities in the Loess Plateau but increased the microbial biomass nitrogen and also increased the Chao1 of bacterial communities related to key functional genes involved in nitrogen cycling. As different crop rotation plans have different effects on STN and PON content, introducing peas or corn into crop rotation could have completely different regulatory effects on the Shannon index of nitrogen cycling key functional genes. Specifically, compared with wheat monoculture, the corn-based crop rotation reduced the Shannon index of key functional genes involved in the nitrogen cycle; however, the pea-based crop rotation significantly increased the Shannon index of these genes compared to the corn-based crop rotation.

Therefore, in order to enhance the ecological potential of soil nitrogen cycling and sustain agricultural development in the Loess Plateau region, it is recommended to introduce leguminous crops into the crop rotation plan.

**Supplementary Materials:** The following supporting information can be downloaded at: https://www.mdpi.com/article/10.3390/agronomy13071769/s1, Table S1. Effects of different planting regimes on the abundance of nitrogen-cycling functional genes (*n* = 3). Values are mean ± standard deviation. Table S2. Effects of different planting regimes on pH, BD and SWC (*n* = 3). Values are mean ± standard deviation. Table S3. Effects of distinct planting regimes on the relative abundance of species related to key functional genes (*n* = 3). Values are mean ± standard deviation. Figure S1. Key functional genes associated with N-cycling processes in different planting regimes. (A) Co-occurrence network of N-cycling functional genes. The circle size represents the degree centrality. (B) Random forest analyses show the significant predictors of functional genes for different nitrogen fractions. MES (%) means the percentage of increase of mean square error. STN, soil total nitrogen; PON, particulate organic nitrogen; PNM, potential nitrogen mineralization; MBN, microbial biomass N; $NH_4+$, ammonium nitrogen; $NO_3-$, nitrate nitrogen; PON/STN, ratio of PON to STN; PNM/STN, ratio of PNM to STN; MBN/STN, ratio of MBN to STN. Figure S2. Relative abundance of the other level classification of species related to key functional genes.

**Author Contributions:** Conceptualization, J.W. and R.L.; methodology, R.L. and F.Z.; software, R.L. and Y.L.; validation, R.L., Y.L. and J.W.; formal analysis, R.L. and Y.L.; investigation, Y.G.; resources, Y.G. and R.L.; data curation, Y.G. and R.L.; writing—original draft preparation, R.L. and F.Z.; writing—review and editing, R.L. and Y.L.; visualization, R.L.; supervision, J.W.; project administration, J.W.; funding acquisition, J.W. All authors have read and agreed to the published version of the manuscript.

**Funding:** This research was funded by the National Natural Science Foundation of China, grant number 42277322, and the Chinese Academy of Sciences "Light of West China", and Shaanxi Agricultural Science & Technology Innovation-Driven Project (NYKJ-2021-XA-005, NYKY-2022-XA-004).

**Data Availability Statement:** All data generated or analyzed during this study are included in this published article.

**Acknowledgments:** The authors thank ChengJie Ren for his help in the determination of soil Metagenomic sequencing of this research.

**Conflicts of Interest:** The authors declare no conflict of interest.

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
