# Peer review of "The Nitrogen Cycling Key Functional Genes and Related Microbial Bacterial Community α−Diversity Is Determined by Crop Rotation Plans in the Loess Plateau"

_agronomy, doi:10.3390/agronomy13071769_

Round 1

Reviewer 1 Report

This is an interesting approach to illustrate and document the effective use of molecular techniques supporting the long time known the significance of crop rotation in crop health and productivity. 

You may refer to the attached sheet of comments and suggestion.

Nitrogen fixing bacteria!!!???

Reviewer 2 Report

Abstract: Should be shortened. The current abstract is 366 words but the guidelines specify it should be 200 words maximum.

128-129: The crop rotation treatment has different crops grown each year. (1) What crops had just been grown on the plots? (2) The information that the samples were taken after harvesting, as described on line 359, should also be mentioned in the methods. (3) Why was the sampling only performed on a single year? I would assume that the crop rotation treatments will give different results depending on which crop is grown that year. If the experiment has been running since 1984, there must be other samples which could have been analysed and will give a much better idea of the differences between the treatments, especially at different stages of crop rotation.

Minor issues with the spacing between numbers and units on lines 108, 133 and 137. Possibly on others too. It should be checked thoroughly.

159-160: No accession number is provided for the data.

168: The equation is incorrectly formatted.

169-170: How was annotation performed?

170-173: (1) The previous study is not cited. (2) Shouldn’t nitrogen fixation genes also be included in the study?

176-185: I don’t understand the motivation behind the network approach and feel some justification should be provided. The biochemical pathways for the different processes are known. Why create a network instead of using the biochemistry to choose the essential, key genes? For example, amoC is chosen as a key gene but it functions as a complex with amoA and amoB. Biologically, it seems that they should function together and any differences in the network and random forest are artefacts of the sequencing and assembly pipelines.

191-197: This seems to be a repeat of section 2.6. The two sections should be merged together.

230-231: Is the Shannon index a meaningful metric for a subset of genes from a metagenome? Is this just calculated from the abundances of the key genes or does this take transcript variants into account?

244-246: How were significant differences between the microbial communities calculated?

247-249: Can you really say that you detected 546 microbial species? Even with 16S amplicon sequencing, identification is usually only to the genus level. In metagenomic work, the databases are less complete and the sequences are less suitable for distinguishing taxa. The best hit matches for sequences are almost certainly not the actual species present.

Figure 3: Why display this at the phylum level if you are confident that you have species level taxonomy? A more detailed breakdown would be good to see and should reveal some differences between the different cropping regimens. Other taxonomic levels could be provided in the supplementaries.

264-265 & 271-273: You show here that there is a positive correlation between the key functional genes and nitrogen fractions but this is not so much a finding as a restatement of your method. You already chose the key functional genes by selecting those which showed explanatory power and a correlation with the various nitrogen factors (see section 2.6 and lines 196-197). Given that, it’s not a surprise that they are correlated, the result couldn’t be anything else.

Figure 4: (1) The colours used for the different cropping rotations should match those used in the previous figures. (2) The Y-axis labels are reversed.

405-408: This conclusion is not supported by the data! You claim here that crop rotations can “enhance the microbial community richness and the abundance of key functional genes” but Figure 2A shows no statistically significant difference between the overall abundance of key functional genes between wheat monoculture and crop rotation. In Table 1, three genes shows significant differences between conditions but amoC and nasA abundances are not statistically different between the crop rotation and monoculture conditions. NasB gives a complicated result with CWWM (not statistically different to F) and PWWM (not statistically different to W) having the lowest and highest abundance respectively. Microbial richness is similarly complicated (Table 2); there is no statistical difference between the crop rotation conditions and the wheat monoculture for the ACE metric. PWWM, W and F are also statistically the same. In Chao1, PWWM and F do not show statistically significant differences.

408-410: According to Figure 1, neither STN nor PON content are significantly different between W and PWWM conditions.

410-413: In Figure 2B there is no significant difference between W and PWWM in terms of the Shannon index for key functional genes.

I am also concerned that several citations appear incorrect. For example, reference 40 is cited to support the use of MetaGeneMark. That reference leads to a 2008 paper which does not appear to perform metagenomics at all and MetaGeneMark seems to only have been released two years later. Reference 5 is cited to support the use of Megahit in the methods but is neither the paper that introduces Megahit nor does it even use the software itself! The methodological citations should be the easiest to include and if these are incorrect, it’s hard to trust that the other reference support their claims either.

Round 2

Reviewer 1 Report

Thank you for your closely dealing with points raised. The last part in your response made it much clearer to me. So, I am sure you may make as well for future readers of your manuscript too.

Reviewer 2 Report

I am very glad to see the authors responded so well to my suggestions. I think this version of the manuscript is much better and, even when the suggestions were not taken, I understand the authors’ reasoning behind their approach.